# Notes from the 2022 Folate, Vitamin B12, and One-Carbon Metabolism Conference

**DOI:** 10.3390/metabo13040486

**Published:** 2023-03-28

**Authors:** Adam G. Maynard, Boryana Petrova, Naama Kanarek

**Affiliations:** 1Department of Pathology, Boston Children’s Hospital, Boston, MA 02115, USA; 2Graduate Program in Biological and Biomedical Sciences, Harvard Medical School, Boston, MA 02115, USA; 3Harvard Medical School, Boston, MA 02115, USA; 4Broad Institute of Harvard and MIT, Cambridge, MA 02142, USA

**Keywords:** folate, vitamin B12, vitamin B9, folic acid, one-carbon metabolism, vitamins, nutrition

## Abstract

Here, we present notes from the Folate, Vitamin B12, and One-Carbon Metabolism Conference organized by The Federation of American Societies for Experimental Biology (FASEB), held in Asheville, North Carolina, USA, 14–19 August 2022. We aim to share the most recent findings in the field with members of our scientific community who did not attend the meeting and who are interested in the research that was presented. The research described includes discussions of one-carbon metabolism at the biochemical and physiological levels and studies of the role of folate and B12 in development and in the adult, and from bacteria to mammals. Furthermore, the summarized studies address the role of one-carbon metabolism in disease, including COVID-19, neurodegeneration, and cancer.

## 1. Introduction

One-carbon (1C) metabolism is centered around two essential vitamins: folic acid (B9, folate) and cobalamin (B12). The function of 1C metabolism is to transfer a one-carbon unit, such as a methyl or formyl group, from donors (i.e., serine) to acceptors (e.g., nucleotide intermediates or methylation targets). As depicted in Figure 1, 1C metabolism can be divided into two interdependent circular sets of reactions, namely the folate and methionine cycles. The folate cycle plays an essential role in nucleotide synthesis, as 10-formyl tetrahydrofolate (10-formyl THF) is utilized in purine synthesis, and 5,10-methylene THF is used in thymidylate synthesis. The folate cycle can also donate methyl groups to the methionine cycle. These two cycles converge at the B12-dependent enzyme methionine synthase (MTR). This reaction allows 1C units from the folate cycle to enter the methionine cycle and regenerate methionine. Subsequent S-adenosylmethionine (SAM) is an important cellular methyl donor that facilitates the methylation of DNA, RNA, and proteins and is essential for biosynthesis of various small molecules such as polyamines [1].

One-carbon metabolism has been extensively studied over the past several decades due to its direct link to cancer, and because it is a target for anticancer drugs. The historical connections have been reviewed elsewhere [2,3,4]. Here, we highlight the progress in understanding 1C metabolism, as presented at the most recent summer 2022 FASEB 1C metabolism conference. The presentations covered 1C metabolism in a variety of topics, ranging from biochemistry to disease pathology, and spanned research in diverse organisms, from bacteria to humans. Clinically oriented talks are not discussed in this report because these are beyond the focus of this summary, that is, basic-science-oriented, and beyond the expertise of the writers.

## 2. Basic Biochemistry of 1C Metabolism

Several talks focused on fundamental questions in 1C metabolism at the cellular and molecular levels. At the molecular level, Paola Drago (Dublin City University, Dublin, Ireland) presented work on dihydrofolate reductase (DHFR) regulation. DHFR is a key enzyme in folate metabolism because it reduces folic acid to dihydrofolate (DHF) and DHF to tetrahydrofolate (THF). THF is the form of folate that functions as an enzymatic cofactor by receiving, carrying, and donating 1C units for downstream reactions. Paola’s studies on DHFR expression and activity revealed a potential regulatory mechanism through the pseudogene DHFR2.

At the cellular level, Darren Walsh (Dublin City University, Dublin, Ireland) discussed the relationship between 1C metabolism and mitochondrial heteroplasmy, the total mitochondrial DNA (mtDNA) mutations among all mtDNA copies in a cell. He found that dietary and genetic mouse models of B12 deficiency increased mitochondrial heteroplasmy. His work stressed the need to further probe the connection between 1C metabolism and mtDNA damage, its potential functional outcome, and the relevance to human biology. Martha Field (Cornell University, Ithaca, NY, USA) presented related findings, investigating mutations in mtDNA using genetic mouse models of SHMT2 and MTR deficiency. SHMT2-deficient mice had impaired mitochondrial function, and she showed that uracil accumulates in mtDNA upon reduced SHMT2 expression. The combination of folate deficiency and SHMT2 deficiency was neither additive nor synergistic in inducing uracil incorporation [5]. This is contrary to Martha’s findings in an MTR-deficient model (MTR+/−), where folate deficiency alleviated the uracil incorporation induced by the MTR deficiency [6]. Further studies are needed to resolve the underlying mechanistic differences.

## 3. Whole-Body Folate Homeostasis

Many gaps remain in our understanding of how folate is balanced throughout the body. It is not known how and if tissues communicate folate depletion or excess at the whole-body level. Additionally, it has not been determined whether different tissues experience various consequences to 1C-unit deprivation or excess. For example, it is not clear why the nutritional deprivation of folate first clinically manifests as anemia. Adam Maynard (Boston Children’s Hospital, Boston, MA, USA) addressed this by studying the cellular response of erythroid cells to folate deprivation. He found that folate deprivation induced the aberrant and premature differentiation of erythroid cells. His findings were consistent both in cancer cell lines and in primary murine erythroid progenitor cells.

Katarina Heyden (Cornell University, Ithaca, NY, USA) investigated how excess dietary folate combined with B12 deficiency impacted tissue folate levels. Her group utilized the MTR+/− mouse strain mentioned above as a model for B12 deficiency. Katarina found inconsistent changes in folate levels across tissues following feeding with excess folic acid. For example, a diet with excess folic acid resulted in increased plasma folate, but liver and brain folate levels did not change. This work underscores how much there is still to be learned about folate mobility across tissues. Further work is needed to better understand the tissue-specific regulatory processes in response to dietary folic acid perturbations, and the possibility of interorgan folate sharing.

## 4. One-Carbon Metabolism and COVID-19

Another important aspect of whole-body folate homeostasis is the interaction between 1C metabolism and viral infection. Benjamin Gewurz (Brigham and Women’s Hospital, Boston, MA, USA) kicked off the 2022 Folate meeting by sharing his recent work on 1C metabolism and COVID-19. During viral infection and amplification, viral RNA becomes dominant over host RNA. However, the mechanism supporting this excess metabolic burden was unknown. Benjamin Gewurz’s keynote lecture addressed one mechanism used by the SARS-CoV-2 virus to support its transcription-hijacking host 1C metabolism [7]. In addition to identifying the key metabolic pathways altered during infection, including 1C metabolism, this work also tested the utility of 1C metabolism inhibitors as a means of preventing viral replication. The Gewurz group found that inhibition of 1C metabolism in infected cells, either genetically, by inhibition of the enzyme SHMT2, or pharmacologically, with the antifolate MTX, results in a reduced infection rate through reduced virion production. This important connection can aid in the identification of COVID-19 vulnerabilities, as well as future coronaviruses, and augment current antiviral therapies. We would also like to highlight Ralph Green’s (University of California, Davis, Sacramento, CA, USA) talk on the association between folic acid, methotrexate use, and COVID-19. While work from the Gewurz lab highlighted a connection between 1C metabolism and COVID-19 at the molecular level, Ralph Green’s work utilized a UK Biobank cohort to investigate an epidemiological connection [8]. This work uncovered a positive association of folic acid supplementation with COVID-19 diagnosis and COVID-19-associated death. In contrast, this association was not observed in cases that were treated with a combination of folic acid and methotrexate supplementation. This may suggest a protective role for methotrexate, paralleling the findings presented by Benjamin Gewurz. These two studies complement each other in supporting the relationship between 1C metabolism and the viral lifecycle. This opens a potential avenue for modulating folate metabolism to support antiviral therapy.

## 5. One-Carbon Metabolism and Cancer

One-carbon metabolism has been linked to cancer progression and targeted in cancer treatments since the 1940s [9]. Despite decades of research on the role of 1C metabolism in cancer, exciting discoveries are still being made today.

### 5.1. Dependency of Cancer Cells on Methionine and the Methylation Cycle

Several speakers at the meeting presented data regarding the methylation cycle and its role in cancer. Relevant topics were tumor initiation and tumor initiating cells (TICs), cancer growth and progression, and cancer immune modulation through the role of T cells.

Man Hsing Hung (National Cancer Institute, Bethesda, MD, USA) discussed the role of MAT2A, a methionine cycle gene, in cancer from the perspective of T-cell metabolism. Elevated MAT2A expression and methionine cycle metabolism were found to contribute to impaired T-cell effector function, compromised tumor immunity, and increased tumor growth [10]. Zhenxun Wang (Genome Institute of Singapore, Singapore) also investigated the role of MAT2A but from the perspective of tumor initiation, with a special focus on the metabolic dependency of TICs [11]. This work identified elevated methionine cycle activity as a dependency in TICs and revealed a role for elevated MAT2A in mediating this dependency. In another model of tumor initiation, Fares Namour (University of Lorraine, Lorraine, France) compared the methionine dependency of glioblastoma TICs cultured in monolayer vs. spheroid form. Glioblastoma TIC spheroids were shown to be methionine-dependent, while cells grown in a monolayer were methionine-independent. Mechanistic work revealed that knockout of the 1C metabolism mitochondrial enzyme ALDH1L2 increased 1C metabolism and reverted the methionine dependency of spheroid cells.

Madeline Hall (UNC Chapel Hill, Chapel Hill, NC, USA) reported studies in mice with genetically targeted glycine N-methyltransferase (GNMT), another methionine cycle gene. This knockout presents perturbed methylation metabolism due to elevated levels of S-adenosyl-methionine (SAM) and results in hepatocellular carcinoma (HCC). She showed that methionine restriction prevented HCC and reversed the metabolic aberrations caused by GNMT deletion [12]. Caitlin Zacharias (McGill University, Montreal, Quebec, Canada) presented work on the dependency of melanoma cells on methionine through the metabolism of cobalamin-associated C (MMACHC), an enzyme responsible for converting dietary cyanocobalamin to cobalamin (B12).

These studies, among others, have raised the interest in methionine restriction as a possible beneficial dietary intervention in cancer [13]. Interestingly, the benefits of methionine restriction are not only limited to cancer, as Jason Locasale (Duke University, Durham, NC, USA) presented how methionine restriction could have several positive effects on humans (i.e., antiaging and antiobesogenic) [14].

### 5.2. B Vitamins and Their Role in Cancer Progression

Work presented by Sergey Krupenko (University of North Carolina Chapel Hill, Chapel Hill, NC, USA) provided details on the structure and function of ALDH1L1, a cytosolic enzyme that converts 10-formyl THF to THF while also converting NADP+ to NADPH [15]. This enzyme is silenced in tumors, suggesting a possible role as a tumor suppressor. Sergey characterized a functional role for the tetrameric state of ALDH1L1, and showed that knockout of ALDH1L1 promotes tumor progression, providing corroborating evidence that ALDH1L1 is a tumor suppressor.

Ana Gomes (Moffit Cancer Center, Tampa, FL, USA) discussed the role of methylmalonic acid (MMA), a toxic metabolite that accumulates in the context of B12 deficiency, in promoting tumor progression and metastasis. She found that MMA, one of the metabolites elevated in serum from older patients, is sufficient for promoting breast cancer metastases through metabolic and transcriptional modifications that are pro-EMT (epithelial-to-mesenchymal transition). Furthermore, using several genetic perturbations in a triple-negative breast cancer (TNBC) model, Ana showed a strong association between MMA levels and the metastatic spread of TNBC [16]. The role of another essential vitamin, B6, in cancer progression was discussed in a talk by Lingbo Zhang (Cold Spring Harbor Laboratory, Cold Spring Harbor, NY, USA). A functional genetic screen identified PDXK, the enzyme that converts pyridoxal to pyridoxal 5-phosphate (PLP, B6) as a growth dependency in acute myeloid leukemia [17]. CRISPR knockout of PDXK revealed a strong growth inhibition specific to leukemia cells, which was not observed for bone marrow hematopoietic stem cells. This dependency on PDXK and thus B6 is striking as the block in proliferation is similar to that observed for Bcl2, a known AML target.

These studies highlight the well-established notion that B vitamins benefit tumor cells. Although B vitamins are also essential for normal cells, their unique contribution to cancer progression requires holistic consideration in cancer patients’ diet. We should carefully take into consideration that some cancer patients may consume unsupervised multivitamin pills, and evaluate appropriate nutritional recommendations in the future.

## 6. One-Carbon Metabolism and the Microbiome

Folate and B12 cannot be synthesized in animal cells and must be acquired through the diet. Folate can be produced in prokaryotes and some eukaryotes; however, B12 is unique in that it is only synthesized by prokaryotes. Thus, there are B12-directed enzymes that have unexplored functions and properties. B12 is a member of the diverse corrinoid family of compounds, which encompasses many noninterchangeable structures. These compounds can be utilized as cofactors in a variety of enzymatic reactions with some specificity, i.e., enzymes that are adapted to one corrinoid may or may not be able to utilize a different corrinoid. The talks summarized below underscore some of the vast biochemical diversity observed in prokaryotes and outline some possible avenues for how B12 prokaryotic reactions can be co-opted to serve relevant applications for human health.

Bacteria utilize B12 and B12 derivatives in a variety of reactions. In bacteria, metabolic functions often take place in microcompartments: self-organizing organelles comprising enzymes with related functions [18]. Martin Warren (Quadram Institute Bioscience, Norwich, UK) presented on the role of B12 in the propanediol utilization microcompartment. B12 is required for the enzymatic degradation of 1,2-propanediol in a propanediol microcompartment. *Bacteroides* use a special system to acquire adequate B12 levels, to support their growth in the propanediol microcompartment. As shown in bacterial culture, extracellular vesicles expressed by the bacteria scavenge environmental B12 to increase its compartment-specific concentrations, through which they facilitate 1,2-propanediol degradation and usage by the bacteria [19]. Michiko Taga (University of California, Berkeley, Berkeley, CA, USA) demonstrated the potential of corrinoid specificity in modulating the composition of bacterial communities, by selecting for, or against, bacteria that can or cannot utilize specific corrinoids. This might allow for the selection of beneficial bacteria species by modulating the corrinoid microenvironment in any bacterial ecosystem, whether that be the human gut or the soil. Joseph Krzycki (The Ohio State University, Columbus, OH, USA) presented on the bacterial enzyme trimethylamine methyltransferase, MttB, a member of the MttB superfamily. The enzyme MttB is a corrinoid containing enzyme that can demethylate quaternary amines [20]. This work revealed the potential of the MttB family in demethylating trimethylamine (TMA), or TMA precursors, to prevent trimethylamine N-oxide (TMAO) production. TMAO levels correlate with risk of atherosclerosis, diabetes, and thrombosis—these findings highlight the importance of studying bacterial 1C metabolism as it has direct implications for human health.

Another B12-dependent methyltransferase, methanogenesis marker protein 10 (Mmp10), was discussed during a presentation by Olivier Berteau (Université Paris-Saclay, INRAE, Jouy-en-Josas, France). Olivier reported on some new insights learned from structural studies of radical SAM enzymes. Radical SAM enzymes are a family of metallo-protein enzymes with radical substrates, and some require B12 for their function. These are the only enzymes that can bind a methyl group to carbon. Olivier showed that some enzymes from this family use radical SAM both as the methyl donor and as an activator of the enzyme. By applying X-ray absorption spectroscopy, Olivier and colleagues solved the structure of the studied enzymes and exposed two distinguished binding sites for two SAM molecules–one functions as the activator, and one functions as the substrate. Alhosna Benjdia (Université Paris-Saclay, INRAE, Jouy-en-Josas, France) focused on another B12-dependent radical SAM enzyme, tryptophan 2C methyltransferase (TsrM). This enzyme was found not to be radical-based. Additionally, the mechanism by which B12 was able to mediate the methyl transfer depended on an Fe-S center. While this work focused on TsrM’s canonical role in methylating tryptophan, TsrM can also catalyze the methylation of a variety of metabolites. Specifically, TsrM’s ability to methylate inaccessible carbons on benzyl-ring-containing compounds may be useful for designing future biosynthetic reactions that require otherwise challenging methylation steps [21].

In summary, this body of work pointed to multiple avenues where increasing understanding of intricate bacterial metabolism could lead to clinically relevant applications and solutions to benefit human health.

## 7. One-Carbon Metabolism and Neuronal Biology

### 7.1. One-Carbon Metabolism during Development

Nutritional folate deficiency during pregnancy was the leading cause of neural tube defects (NTDs) until folate fortification and dietary supplementation for pregnant women became a broad recommendation [22]. However, we still lack the mechanistic understanding and the molecular etiology of this strong causal relationship between folate and proper development. Several talks addressed this mechanistic gap.

Nicholas Greene (University College London, London, United Kingdom) studied the role of the mitochondrial 1C metabolism enzyme glycine decarboxylase (GLDC) in neural tube closure. GLDC deficiency results in NTDs, and this work showed that this is due to the inhibition of mitochondrial 1C-derived formate production. Indeed, formate supplementation rescued GLDC loss-mediated NTDs [23]. Interestingly, Kit-Yi Leung (University College London, London, UK) showed that the postnatal disease caused by GLDC loss, nonketotic hyperglycinemia (NKH), can be rescued by approaches that reduce glycine levels in the plasma (and, as a consequence, also in the brain) [24]. These approaches include treatment with benzoate, cinnamate (a glycine-conjugation intermediate), and liver targeted GLDC exogenous expression. All of these approaches showed promise in restoring metabolic homeostasis in NKH.

Additional work on NTDs by Ron Parchem (Baylor College of Medicine, Houston, TX, USA) assessed the role of miRNAs in regulating 1C metabolism. miR302 is expressed during neural tube closure and is essential for the prevention of NTDs [25]. In parallel to increasing NTDs, knockout of miR302 led to extensive metabolic changes in the developing brain and provided more evidence for the importance of miRNAs in regulating metabolism.

Another well-known link between 1C metabolism and developmental disease is the in-born error disease methylmalonic acidemia (MMA). MMA is characterized by the build-up of methylmalonic acid, a toxic metabolite that forms due to B12 deficiency or mutations in the B12 enzyme methylmalonyl-CoA mutase (MUT). Charles Venditti (National Institutes of Health, Bethesda, MD, USA) employed the exogenous expression AAV in vivo system and genome editing to treat this disease. He presented promising results on the exogenous expression of the wildtype MUT in mice, with the hope of decreasing plasma MMA levels. This work is encouraging and might open an avenue for using gene therapy to balance patient metabolism in the future [26].

While antifolate’s role in neurological impairment is well documented, Konstantinos Zarbalis (University of California, Davis, CA, USA) shared work on the surprising and worrying consequences of excess folic acid supplementation during pregnancy in the embryo’s brain. His work in mice revealed that excess folic acid altered embryonic cortical neurogenesis and led to behavioral abnormalities [27]. This suggests a negative consequence of excess folic acid during development. Indeed, Karen Christensen (McGill University, Montreal, QC, Canada) also reported negative consequences of excess folic acid on embryonic development; she showed developmental delays caused by excess folic acid and revealed differences in 1C metabolism based on mouse strain and sex [28]. As an example, Mthfr−/− male mice were infertile in the BALB/c background and fertile in the C57BL/6J strain, emphasizing the importance of considering these differences when designing and interpreting 1C metabolism studies. Natasha Bobrowski-Khoury (SUNY Downstate Medical Center, Brooklyn, NY, USA) continued the discussion on 1C metabolism during pregnancy and behavior defects in the embryo. Her work first demonstrated that folate receptor alpha (FRα) autoantibodies were present in the majority of women who gave birth to fetuses with neural tube defects. Then, work conducted in rats revealed a direct connection between FRα antibodies and behavioral deficits, with these traits persisting across generations. Another aspect of folate-related neural tube defect was presented by Joydeep Chakraborty (Texas A&M University, TX, USA), who studied peripheral neuropathy, a condition that causes loss of sensation and is associated with neural tube defects [29]. Dr. Chakraborty presented findings using a mouse model of *Shmt* loss. SHMT is a major provider of one-carbon groups for all folate-dependent biosynthetic reactions (Figure 1). The *Shmt1*-null mice were viable and sometimes presented neural tube defects that could be rescued by folic acid supplementation. Electrophysiological experiments indicated a significant *Shmt1*-depletion-driven defect in peripheral neurons comparable to an established neuropathy model based on the Lepr^db/db^ diabetic mouse. Interestingly, SHMT1 effects were observed in female but not male mice. In females, these effects were rescued using either deoxyuridine or folic acid supplementation.

These studies provide a strong incentive to continue investigation into both insuffienct and excessive folic acid intake during pregnancy, as both folate deprivation and surplus can result in neuronal deficits in the developing brain.

### 7.2. Postnatal Neuronal Defects Related to B12 Deficiency

David Coelho (University of Lorraine, Nancy, France) presented data on perturbation of methionine synthesis by *Mtr* deletion in mouse neurons to simulate B12 deficiency (Figure 1). *Mtr* knockout led to significant behavioral, learning, and memory delays, as well as ER and oxidative stress [30]. These negative associations could be reversed by increasing *Sirtuin1* (SIRT1) activity, which increased deacetylation of heat shock factor 1 (HSF1) and reduced ER stress. This suggests a potential mechanism for B12-induced neurological impairment and a clinically relevant potential approach for treating B12-related in-born errors. Rosa-Maria Gueant-Rodriguez (University of Lorraine, Nancy, France) presented a study on methionine synthase during neurogenesis at postnatal day 21. Knockout of *Mtr* led to increased neuronal glycolysis, mitochondrial dysfunction, and oxidative respiration. This manifested through impaired differentiation of neural stem cells that resulted in decreased numbers of neurons and increased numbers of astrocytes. This work revealed another mechanistic association between 1C metabolism defects and neural development and function. In another talk by Rosa-Maria Gueant-Rodriguez, the role of MTR in the eye was studied using a brain- and eye-confined genetic knockout. This work revealed a connection between methionine synthase activity, decreased SAM/SAH, decreased opsin, and decreased cone cells. It is still not clear if the MTR-deficiency-induced visual acuity is the result of reduced proliferation or increased apoptosis of the cone cells. Jessica Tanis (University of Delaware, Newark, DE, USA) presented work on the connection between B12 metabolism and Alzheimer’s disease progression using *C. elegans* as a model organism. In this work, she revealed that dietary B12 reduced oxidative stress, mitochondrial impairment, and amyloid beta accumulation [31].

These mechanistic studies demonstrate the impact of B12-related neuronal pathology and underline the power of molecular research and animal models in metabo-neurobiology.

### 7.3. One-Carbon Metabolism and Neuronal Biology in the Adult

In addition to talks focused on the role of 1C metabolism in neurological function from the developmental perspective, Boryana Petrova (Boston Children’s Hospital, Boston, MA, USA) looked at neuronal biology in the adult. She presented on a new strategy to minimize toxic neurological side effects in pediatric patients treated with the antifolate chemotherapy methotrexate. Methotrexate is directly delivered to the cerebrospinal fluid (CSF) to kill leukemia cells colonizing the brain; however, this leads to neurological side effects [32]. This work demonstrated that methotrexate triggered oxidative stress in the CSF, which was subsequently propagated to neuronal tissues such as the hippocampus. Methotrexate-induced oxidative stress was counteracted by the antioxidant gene superoxide dismutase 3 (SOD3) and gene therapy targeted to the choroid plexus. This AAV-based SOD3 overexpression approach reduced neurological impairment in methotrexate-treated mice, as validated by behavioral studies and metabolite profiling of the CSF and neurons.

## 8. One-Carbon Metabolism Novel Pathologies and Beyond

Novel links between 1C metabolism and disease are still emerging. Jae Woo Jung (University of Pennsylvania, Philadelphia, PA, USA) presented data that connected 1C unit utilization and dilated cardiomyopathy (DCM). Metabolite profiling of nonfailing hearts and DCM hearts revealed a number of 1C metabolism alterations, including a decrease in the methylation cycle. Hieronim Jakubowski (Rutgers University, Newark, NJ, USA) studied male fertility in the context of 1C metabolism; specifically, he studied the role of cystathionine beta-synthase (CBS) in murine sexual signaling. CBS removes homocysteine from the methylation cycle and affects 1C metabolism. Hieronim’s work revealed a role for CBS in regulating major urinary protein expression. Inactive CBS, which prevents shunting homocysteine from the methionine cycle, reduced pregnancy and male fertility compared with WT mice and connects impaired metabolism with reproductive fitness [33].

Presenting on an exciting novel function for protein methylation, Lauren Albrecht (University of California, Irvine, CA, USA) reported a link between protein methylation, Wnt signaling, and protein degradation in the lysosome. Through a compound library screen, her laboratory showed that GSK3 lysosomal targeting and subsequent degradation was dependent on di-methylarginine modification. This post-translational modification is dependent on the activity of protein arginine methyltransferase 1 (PRMT1) and the universal methyl donor SAM. Treatments that decreased levels of SAM (such as methotrexate) were potent inhibitors of Wnt signaling and thus presented a potential tool for dietary modification of the Wnt pathway.

We think that novel discoveries linking 1C metabolism and disease will continue to emerge in the future, underscoring an essential role for this central metabolic pathway in human health.

## 9. Concluding Remarks

Through several decades of research, it has become clear that 1C metabolism is intricately connected with many cell processes to maintain cell function, fate, and viability. This manifests strongly during times of folate and/or B12 depletion, when numerous pathologies in various tissues emerge. These pathologies provide a strong motivation to research 1C metabolism.

While many important clinical associations and mechanistic studies have been made, it is clear that we have much more to achieve at the basic science level to fully understand, characterize and explain these clinical observations in order to be able to offer targeted and effective treatments. The work presented here serves to summarize the substantial progress that has been made over the past several years to understand the great mysteries in 1C metabolism. We would like to highlight two exciting new trends that emerged from this meeting. First, the novel role of 1C metabolism in COVID-19 has emerged. It was shown that the virus SARS-CoV-2 hijacks this pathway and harnesses it to support viral propagation. We left with the strong impression that further studies will expand our understanding of the dependency of the SARS-CoV-2 virus, and likely other common respiratory viruses, on 1C metabolism and that such research will result in novel treatment strategies. The second strong trend was the importance of maintaining an adequate, but not excessive, level of folic acid consumption. In addition to the wide body of literature on folic acid deficiency associated pathologies, we heard from several speakers about the adverse effects of high-dose folic acid supplementation. Clearly, we should continue to study the basic cellular and systemic physiological consequences of both high and low folic acid intake in order to better inform the healthy fortification and supplementation of folic acid.

We are awaiting and thrilled about the research that will be performed in the coming years in the 1C metabolism field to follow up on the exciting reports we summarized here.

## Figures and Tables

**Figure 1 metabolites-13-00486-f001:**
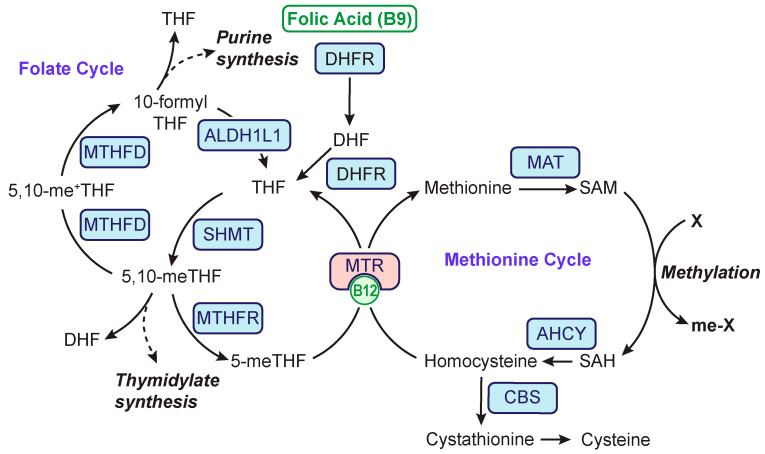
**Schematic of one-carbon metabolism**. The scheme includes the cytoplasmic folate and methionine cycles. B12 denotes vitamin B12. X denotes a methylation target. Abbreviations: MTHFD—methylenetetrahydrofolate dehydrogenase; ALDH1L1—aldehyde dehydrogenase 1 family members L1; SHMT—serine hydroxymethyltransferase; MTHFR—methylenetetrahydrofolate reductase; DHFR–dihydrofolate reductase; MTR—methionine synthase reductase; MAT—methionine synthase; AHCY—adenosylhomocysteinase; CBS—cystathionine β-synthase; B12—vitamin B12; DHF—dihydrofolate; THF—tetrahydrofolate; 5-me-THF—5-methyl-THF; 5-me+-THF—5-methylene-THF; SAM—S-adenosyl methionine; and SAH–S-adenosylhomocysteine.

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
