# Peer review of "Notes from the 2022 Folate, Vitamin B12, and One-Carbon Metabolism Conference"

_metabolites, 2023, doi:10.3390/metabo13040486_

Round 1
Reviewer 1 Report
The manuscript provides the information regarding the studies and discussions presented at the 2022 FASEB Summer Research Conference from the "Folate, Vitamin B12 and One-Carbon Metabolism" series. The authors summarized five days of presentations on basic aspects of One-Carbon Metabolism and highlighted the novel developments in the field. The importance of the area of research and presented topics are covered adequately.
There are several minor edits that could improve the manuscript.
1. Line 32: "due to its direct link to cancer, and it being a target for anti-cancer drugs"
2. Line 37: If readers are referred to the summary of clinical studies, there should be a reference or at least mention for where to find it.
3. The authors might want to make adjustments to the schematic of one-carbon metabolism on figure 1. In the current form, it looks like 5,10-meTHF and 5,10-me+THF are leaving the folate cycle to produce thymidylate and pyrimidines, while it is only one-carbon group that leaves, the remaining cofactors (DHF and THF) remain in the folate pool.
4. Line 44: enzyme abbreviation should be matching to the schematic: ALDH1L1/2 -Aldehyde Dehydrogenase 1 Family Members L1 and L2.
5. Lines142-143: better to re-phrase as: "MH ... reported studies in mice with genetically targeted glycine N-methyl transferase (GNMT), another methionine cycle gene."
6. Lines 198-199: the concept of "propanediol utilization microcompartments" needs to be better explained as the term is used repeatedly. Is this the part of microbiome?
7. Lines 202-204: Needs to be re-phrased for clarity. "Extracellular vesicles..." - from which cells and where (in culture, or gut, or somewhere else?).
8. Line 292: It may be more accurate to state: "SHMT is a major provider of one-carbon groups for all folate-dependent biosynthetic reactions", since the one-carbon group obtained from serine can be reduced or oxidized (depending on cell needs) to variety of other groups and used for purine or pyrimidine biosynthesis, methylation or other reactions.
9. Line 360: Clarification needed regarding "...subsequent degradation was dependent on 5-methyl-arginine methylation." Is process dependent on Arg methylation to mono-methyl-Arg, or on the second methylation to di-methyl-Arg?
10. Line 380: "role of 1C metabolism in COVID-19 was explored" may sound better as "role of 1C metabolism in COVID-19 has emerged..."
11. Line 392: may be, re-phrase to " We are awaiting and thrilled for the research..."
Author Response
Reviewer 1:
The manuscript provides the information regarding the studies and discussions presented at the 2022 FASEB Summer Research Conference from the "Folate, Vitamin B12 and One-Carbon Metabolism" series. The authors summarized five days of presentations on basic aspects of One-Carbon Metabolism and highlighted the novel developments in the field. The importance of the area of research and presented topics are covered adequately.
There are several minor edits that could improve the manuscript.
Thank you for reviewing our paper!
We have addressed your comments and edited our manuscript accordingly.
- Line 32: "due to its direct link to cancer,and itbeing a target for anti-cancer drugs"
Thank you for noticing this error. We have now corrected it.
- Line 37: If readers are referred to the summary of clinical studies, there should be a reference or at least mention for where to find it.
Following personal communication with Dr. Gueant, we now learned that this summary will not be published. We therefore changed our statement and wrote:
Clinically oriented talks are not discussed in this report because these are beyond the focus of this summary, that is basic science-oriented, and beyond the expertise of the writers.
- The authors might want to make adjustments to the schematic of one-carbon metabolism on figure 1. In the current form, it looks like 5,10-meTHF and 5,10-me+THF are leaving the folate cycle to produce thymidylate and pyrimidines, while it is only one-carbon group that leaves, the remaining cofactors (DHF and THF) remain in the folate pool.
We have now updated the figure and so it depicts DHF and THF as products of thymidylate and purine synthesis, respectively.
Please see the attached file for the figure
- Line 44: enzyme abbreviation should be matching to the schematic: ALDH1L1/2 -Aldehyde Dehydrogenase 1 Family Members L1 and L2.
Thank you for noticing this error. We have now corrected it.
- Lines142-143: better to re-phrase as: "MH ... reported studies in mice with genetically targeted glycine N-methyl transferase (GNMT), another methionine cycle gene."
Thank you for the great suggestion. We have now corrected it accordingly.
- Lines 198-199: the concept of "propanediol utilization microcompartments" needs to be better explained as the term is used repeatedly. Is this the part of microbiome?
We added a clarification of the term and a reference for additional reading:
In bacteria, metabolic functions often take place in microcompartments: self-organizing organelles comprised of enzymes with related functions [17].
- Lines 202-204: Needs to be re-phrased for clarity. "Extracellular vesicles..." - from which cells and where (in culture, or gut, or somewhere else?).
We have clarified this sentence and added the missing information, as well as citation too the recently-published paper describing these results:
As shown in bacterial culture, extracellular vesicles expressed by the bacteria scavenge environmental B12 to increase its compartment-specific concentrations and by that facilitate 1,2-propanediol degradation and usage by the bacteria.
- Line 292: It may be more accurate to state: "SHMT is a major provider of one-carbon groups for all folate-dependent biosynthetic reactions", since the one-carbon group obtained from serine can be reduced or oxidized (depending on cell needs) to variety of other groups and used for purine or pyrimidine biosynthesis, methylation or other reactions.
Great point! We have replaced the sentence with the one suggested by the reviewer.
- Line 360: Clarification needed regarding "...subsequent degradation was dependent on 5-methyl-arginine methylation." Is process dependent on Arg methylation to mono-methyl-Arg, or on the second methylation to di-methyl-Arg?
To address this comment, I consulted with Dr. Albrecht, who rerecommended replacing “5-methyl-arginine methylation” with “di-methylarginine modification”. Now the sentence reads:
Through a compound library screen, her lab showed that GSK3 lysosomal targeting and subsequent degradation was dependent on di-methylarginine modification.
- Line 380: "role of 1C metabolism in COVID-19 was explored" may sound better as "role of 1C metabolism in COVID-19 has emerged..."
We have replaced the sentence with the one suggested by the reviewer.
- Line 392: may be, re-phrase to " We are awaiting and thrilled for the research..."
We have replaced the sentence with the one suggested by the reviewer.

Reviewer 2 Report
This is a great summary of the meeting, which will be of interest for the scientific community. I have only several minor suggestions/edits.
Fig. 1 and the legend: The schematic shows cytoplasmic 1C metabolism while the legend also incorporates some elements from mitochondrial folate metabolism (ETC, SHMT2, ALDH1L2, MTHFD1/2). This needs to be corrected. For clarity, the purine pathway in the schematic should indicate the production of THF.
Lines 12-13, I suggest to re-phrase the sentence as follows: We aim to share the most recent findings in the field with members of our scientific community who did not attend the meeting and who are interested in the research which was presented.
Line 22: As depicted in Figure 1, …
Line 30: Perhaps participation of SAM in the biosynthesis of small molecules should be also acknowledged.
Line 160: NADP should be NADP+.
Line 385: The second
Author Response
Reviewer 2
This is a great summary of the meeting, which will be of interest for the scientific community. I have only several minor suggestions/edits.
We thank the reviewer for this assessment.
Fig. 1 and the legend: The schematic shows cytoplasmic 1C metabolism while the legend also incorporates some elements from mitochondrial folate metabolism (ETC, SHMT2, ALDH1L2, MTHFD1/2). This needs to be corrected. For clarity, the purine pathway in the schematic should indicate the production of THF.
Thank you for pointing this out. We have now corrected both the figure and the legend to depict only the cytosolic folate cycle and added THF and DHF as products of purine and pyrimidine synthesis, respectively:
Please see figure in the attached file.
Figure 1. Schematic of one-carbon metabolism. The scheme includes the cytoplasmic folate and methionine cycles. B12 denotes vitamin B12. X denotes a methylation target. Abbreviations: MTHFD – Methylenetetrahydrofolate dehydrogenase; ALDH1L1 -Aldehyde Dehydrogenase 1 Family Members L1; SHMT – serine hydroxymethyltransferase; MTHFR – methylenetetrahydrofolate reductase; DHFR -dihydrofolate reductase; MTR – methionine synthase reductase; MAT – methionine synthase; AHCY – Adenosylhomocysteinase; CBS – cystathionine β-synthase; B12 – vitamin B12; DHF -dihydrofolate; THF -tetrahydrofolate; 5-me-THF – 5-methyl-THF; 5-me+-THF – 5-methylene-THF; SAM – S-Adenosyl methionine ; SAH – S-adenosylhomocysteine; X denotes a methylation target.
Lines 12-13, I suggest to re-phrase the sentence as follows: We aim to share the most recent findings in the field with members of our scientific community who did not attend the meeting and who are interested in the research which was presented.
We updated this sentence as suggested.
Line 22: As depicted in Figure 1, …
Thank you for noticing this error. We have now corrected it.
Line 30: Perhaps participation of SAM in the biosynthesis of small molecules should be also acknowledged.
This is a great point. We added this information and a relevant reference for further reading:
Subsequent S-adenosylmethionine (SAM) is an important cellular methyl donor that facilitates methylation of DNA, RNA, and proteins, and is essential for biosynthesis of various small molecules, such as polyamines [1].
Line 160: NADP should be NADP+.
Thank you for noticing this error. We have now corrected it.
Line 385: The second
Thank you for noticing this error. We have now corrected it.

Reviewer 3 Report
This is a useful, comprehensive and well-written review of the 2022 FASEB Meeting regarding Folate, Vitamin B12 and One-Carbon Metabolism. I have only a very few minor observations:
a) It perhaps isn't possible at present, but if it is, could the Authors kindly provide a reference to Dr Jean-Louis Gueant's summary of clinically oriented talks pertaining to the Conference (Pg 1, Line 37-38).
b) Should it be "Darren Walsh" not "Daren" (Pg 2, Line 60)
c) MMAHC should be "MMACHC" (Pg 4, Line 149)
Author Response
Reviewer 3
This is a useful, comprehensive and well-written review of the 2022 FASEB Meeting regarding Folate, Vitamin B12 and One-Carbon Metabolism. I have only a very few minor observations:
We thank the reviewer for this assessment.
- a) It perhaps isn't possible at present, but if it is, could the Authors kindly provide a reference to Dr Jean-Louis Gueant's summary of clinically oriented talks pertaining to the Conference (Pg 1, Line 37-38).
Following personal communication with Dr. Gueant, we now learned that this summary will not be published. We therefore changed our statement and wrote:
Clinically oriented talks are not discussed in this report because these are beyond the focus of this summary, that is basic science-oriented, and beyond the expertise of the writers.
- b) Should it be "Darren Walsh" not "Daren" (Pg 2, Line 60)
Thank you for noticing this typo. This is now corrected.
- c) MMAHC should be "MMACHC" (Pg 4, Line 149)
Thank you for noticing this typo. This is now corrected.
Reviewer 4 Report
Here are my personal suggestions to improve the paper:
1-In the abstract, please insert the topic treated e.g 1C metabolism cancer, covid19 etc...
2-Figure 1 the legenda contains an error MTR is not a Methionine synthase, but methionine synthase reductase.
3-line 104, what are the results of this study?
Author Response
Reviewer 4
Here are my personal suggestions to improve the paper:
Thank you for reviewing our paper!
We have addressed your comments and edited our manuscript accordingly.
1-In the abstract, please insert the topic treated e.g 1C metabolism cancer, covid19 etc...
We have now added a description of the summarize research to the abstract:
The research described includes discussions of one-carbon metabolism at the biochemical and physiological levels, studies of the role of folate and B12 in development and in the adult, and from bacteria to mammals. Further, summarized studies address the role of one-carbon metabolism in disease, including covid-19, neurodegeneration, and cancer.
2-Figure 1 the legenda contains an error MTR is not a Methionine synthase, but methionine synthase reductase.
Thank you for noticing this error. We have now corrected it.
3-line 104, what are the results of this study?
We have now added a description of the findings of the described work:
In addition to identifying the key metabolic pathways altered during infection, including folate- and serine-1C metabolism, this work also tested the utility of 1C metabolism inhibitors as a means of preventing viral replication. The Gewurz group found that inhibition of 1C metabolism in infected cells, either genetically, by inhibition of the enzyme SHMT2, or pharmacologically, with the antifolate MTX, results in reduced infection rate through reduced virion production.